# Portable Sensors for Dynamic Exposure Assessments in Urban Environments: State of the Science

**DOI:** 10.3390/s24175653

**Published:** 2024-08-30

**Authors:** Jelle Hofman, Borislav Lazarov, Christophe Stroobants, Evelyne Elst, Inge Smets, Martine Van Poppel

**Affiliations:** 1Environmental Intelligence Unit, Flemish Institute for Technological Research (VITO), Vlasmeer 5, 2400 Mol, Belgium; 2Flanders Environmental Agency (VMM), Kronenburgstraat 45, 2000 Antwerp, Belgiummartine.vanpoppel@vito.be (M.V.P.)

**Keywords:** air quality, sensors, exposure, assessment, citizens, validation

## Abstract

This study presents a fit-for-purpose lab and field evaluation of commercially available portable sensor systems for PM, NO_2_, and/or BC. The main aim of the study is to identify portable sensor systems that are capable of reliably quantifying dynamic exposure gradients in urban environments. After an initial literature and market study resulting in 39 sensor systems, 10 sensor systems were ultimately purchased and benchmarked under laboratory and real-word conditions. We evaluated the comparability to reference analyzers, sensor precision, and sensitivity towards environmental confounders (temperature, humidity, and O_3_). Moreover, we evaluated if the sensor accuracy can be improved by applying a lab or field calibration. Because the targeted application of the sensor systems under evaluation is mobile monitoring, we conducted a mobile field test in an urban environment to evaluate the GPS accuracy and potential impacts from vibrations on the resulting sensor signals. Results of the considered sensor systems indicate that out-of-the-box performance is relatively good for PM (R^2^ = 0.68–0.9, Uexp = 16–66%, BSU = 0.1–0.7 µg/m^3^) and BC (R^2^ = 0.82–0.83), but maturity of the tested NO_2_ sensors is still low (R^2^ = 0.38–0.55, Uexp = 111–614%) and additional efforts are needed in terms of signal noise and calibration, as proven by the performance after multilinear calibration (R^2^ = 0.75–0.83, Uexp = 37–44%)). The horizontal accuracy of the built-in GPS was generally good, achieving <10 m accuracy for all sensor systems. More accurate and dynamic exposure assessments in contemporary urban environments are crucial to study real-world exposure of individuals and the resulting impacts on potential health endpoints. A greater availability of mobile monitoring systems capable of quantifying urban pollutant gradients will further boost this line of research.

## 1. Introduction

Air quality has improved significantly over the past decades. Yet, exposure to particulate matter and nitrogen dioxide in Europe still causes an estimated 253,000 and 52,000 premature deaths per year [1]. Moreover, continuous worldwide urbanization results in megacities with intrinsic hotspots, highlighting the importance of proper air pollution monitoring. Currently, the exposure of the population to air pollution is still determined based on home address (static exposure). However, research has shown that people are exposed to the highest air pollution peaks at times when they are in transit (e.g., during commutes) [2,3,4,5,6]. Studies applying activity-based models or personal monitors demonstrated that transit activities, although short in duration, can be responsible for quite a large part of the integrated personal exposure to combustion-related pollutants [2,4,7,8,9]. Research based on an extensive dataset of 20,000 citizens confirmed that this in-transit (dynamic) exposure is often (64% of the individuals) higher than the respective static residence-based exposure [10]. To better assess dynamic exposure on a wider scale, mobile monitoring systems are needed that (i) can easily be used by study participants (e.g., citizens) and (ii) produce reliable data.

Recent advances in sensor and Internet of Things (IoT) technologies have resulted in a wide range of commercially available “low-cost” sensor systems that allow for quantification of urban pollutants, e.g., particulate matter (PMx), nitrogen dioxide (NO_2_), and ozone (O_3_), at an unprecedented scale [11]. Portable air quality sensors enable quantification of dynamic exposure while raising awareness among citizens about their personal exposure, in turn driving behavioral change [12,13,14,15,16]. Moreover, the obtained mobile data can be used to construct urban exposure maps offering policy makers the right tools for evidence-based policy measures [11,17,18,19,20,21,22]. As Helbig et al. [23] stated, wearable sensing has two aspects: firstly, the exposure of an individual is recorded, and secondly, individuals act as explorers of the urban area. While many stationary sensor systems have been evaluated and benchmarked in previous years [24,25], mobile sensor systems have different requirements, e.g., power autonomy (battery), a high monitoring resolution, and accurate positioning (GPS). Also, the sensor signal noise and between-sensor variability should be low enough to be able to measure the spatial concentration variability at a high temporal resolution (with multiple sensors). Today, many commercially available portable sensor systems are already on the market, but it is hard to determine their fit-for-purpose. This is one of the first studies benchmarking commercially available portable sensor systems for mobile applications. This study includes an evaluation of the data quality performance of different sensor systems under lab and field conditions, as well as during a mobile field test to evaluate GPS performance, the impact of vibrations on the sensor signal, and the overall potential to capture spatial pollutant gradients in urban environments. Doing so, we evaluate the applicability of these sensor systems in real-world urban environments.

## 2. Materials and Methods

### 2.1. Sensor System Selection

Based on an earlier literature market study on air quality sensors [26], expert network consultation (RIVM, VMM, NPL, Ineris, US EPA), reports of independent sensor performance studies (AIRlab, AQ-SPEC, SamenMeten, EPA Air Sensor Toolbox), recent sensor-based citizen science studies [13,27,28,29,30,31,32,33], and a new literature search on Web of Science (~90 publications with search criteria “mobile”, “sensor”, “pollution”, “exposure”), we compiled a longlist of 39 sensor systems with the following criteria:commercially availablewireless/power solution (battery or via car)weatherproof housingdata transmission/logging solution (internal, USB, Bluetooth, LTE-M, LoRA, wifi)

In addition to the criteria above, we defined a set of quantitative (R^2^, slope, intercept, accuracy, and between-sensor uncertainty) and qualitative criteria (price, monitored pollutants, additional variables (temperature, relative humidity, pressure, noise, …), monitoring resolution, GPS localization, autonomy, display/LED, user-friendliness (portability, mounting options, size, weight)) to differentiate between the longlist sensor systems. This longlist was narrowed down based on the following:sensor capability to monitor PM, NO_2_, and/or BCavailability of particle mass concentration (µg/m^3^; instead of particle number concentration)power autonomy (battery instead of car-powered systems)GPS localization (internal or via smartphone)

This resulted in a final shortlist of 12 suitable portable sensor systems for which quotation requests were sent out. Ultimately, 10 sensor systems were purchased (Appendix A), of which 8/10 contained a PM_2.5_ and PM_10_ sensor, and 3/10 sensor systems contained an additional NO_2_ sensor (SODAQ NO_2_, DST Observair, and 2BTech PAM). All 10 sensor systems can be regarded as portable air quality sensor systems, with power autonomy (battery), data storage and/or transmission, and GPS localization (Figure 1).

As no commercial low-cost sensor systems were available for BC, we considered a mid-end instrument, including NO_2_ (DST Observair), and research prototype for stationary measurements (wifi, power cable) in the field co-location campaign. In order to obtain a portable BCmeter, additional hardware/software developments will be needed.

### 2.2. Benchmarking Protocol

The purchased sensor systems were evaluated under controlled (laboratory) and real-life (field) conditions (Figure 2). Field benchmarking included a mobile test on a cargo bike and a 3-month co-location campaign at a regulatory urban background (R801) air quality monitoring station in Antwerp, Belgium.

#### 2.2.1. Laboratory Test Protocol

Laboratory tests were performed for both PM and NO_2_.Test levels and test conditions for NO_2_ were based on the CEN/TS 17660-1:2022. For PM, we included a laboratory test to evaluate the potential of the sensor to measure the coarse fraction (PM_2_._5–10_ = PM_10_ − PM_2_._5_) because it is known that some low-cost sensors calculate PM_10_ concentrations based on the measured concentrations of PM_2_._5_, and sensors can have various response characteristics regarding size selectivity [34,35]. For PM_2_._5_ and PM_10_, we evaluated:Lack-of-fit (linearity) at setpoints 0, 30, 40, 60, 130, 200, and 350 µg/m^3^ (PM_10_, dolomite dust). This concentration range can be considered representative for exhibited PM levels in typical urban environments [2,36,37,38,39,40,41,42]. A Palas Particle dispenser (RBG 100) system connected to a fan-based dilution system and aluminum PM exposure chamber was used.Sensitivity of PM sensor to the coarse (2.5–10 µm) particle fraction. We dosed, sequentially, 7.750 µm and 1.180 µm-sized monodisperse dust (silica nanospheres with density of 2 g/cm^3^) using an aerosolizer (from the Grimm 7.851 aerosol generator) system connected to a fan-based dilution system and an aluminum PM exposure chamber with fans to have homogeneous PM concentrations. This testing protocol is currently considered to be included in the CEN/TS 17660-2 (in preparation) on performance targets for PM sensors.

Based on the lack-of-fit results, the comparability against the reference is evaluated from the resulting linearity (R^2^), accuracy (A; %), Root Mean Squared Error (RMSE), Mean Absolute Error (MAE), Mean Bias Error (MBE), and Expanded Uncertainty (Uexp). As reference instrument, we used a Grimm 11-D with heated sampling inlet line (EDM 264, Grimm). The accuracy is calculated per concentration setpoint as (1) in the lack-of-fit test and evaluated between the sensors as an overall average of all setpoints (mean of means):(1)$$A\;\left(\%\right)=100-\frac{\left|\bar{sensor}-\bar{REF}\right|}{\bar{REF}}\times 100$$

The comparability between the sensors can be regarded as the observed variability between sensors of the same type and is calculated by the between-sensor uncertainty (*BSU* (2)):(2)$${BSU}_{sensor}=\sqrt{\frac{\sum_{i=1}^{n}\sum_{j=1}^{k}{\left({sensor}_{ij}-{average}_{i}\right)}^{2}}{\left(n-1\right)}}$$
with *n*, the number of sensors (3), *k*, the number of measurements over time, ${sensor}_{ij}$, the sensor measurements for period *i*, and ${average}_{i}$, the mean result for period *i*. 

In addition, we calculated the minimal and maximal observed Pearson correlation (r) and MAE (µg/m^3^) between the sensors of the same brand in order to evaluate the intra-sensor comparability.

For NO_2_, we evaluated the following:

Lack-of-fit (linearity) at setpoints of 0, 40, 100, 140, and 200 μg/m^3^.Sensor sensitivity to relative humidity at 15, 50, 70, and 90% (±5%) during stable temperature conditions of 20 ± 1 °C.Sensor sensitivity to temperature at −5, 10, 20, and 30 °C (±3 °C) during stable relative humidity conditions of 50 ± 5%.Sensor cross-sensitivity to ozone (120 µg/m^3^) at zero and 100 µg/m^3^.Sensor response time under rapidly changing NO_2_ concentrations (from 0 to 200 µg/m^3^).

From the lack-of-fit tests, the comparability against the reference was evaluated from the resulting linearity (R^2^), accuracy (%), Root Mean Squared Error (RMSE), Mean Absolute Error (MAE), Mean Bias Error (MBE), and Expanded Uncertainty (Uexp).

In addition, we evaluated sensor stability (mean of exhibited standard deviations at each (stable) concentration setpoint in the lack-of-fit test) and intra-sensor comparability by calculating the between-sensor uncertainty (BSU). As reference instrument, we applied a Thermo Scientific 42iQ-TL chemiluminescence monitor (Thermo Fisher, Waltham, MA, USA).

#### 2.2.2. Mobile Field Test

The mobile field test aimed at testing the GPS accuracy of the sensor systems along a ~10 km trajectory within the varying urban landscape (street canyons, open parks, tunnels, …) of Antwerp, Belgium (Figure 3). GPS accuracy was evaluated by calculating the average horizontal distance (m) of the high-resolution mobile GPS measurements to a reference GPS track. The reference GPS track was determined by evaluating 3 different GPS platforms (TomTom Runner2, Garmin Edge 810, and Komoot smartphone application) and selecting the best performing one as the reference GPS trajectory.

#### 2.2.3. Field Co-Location Campaign

During the field co-location campaign, the considered sensor systems were exposed for a period of 3 months (7 September 2022–5 December 2022) to ambient pollutant concentrations in an actively vented outdoor shelter, deployed on top (near the air inlets) of a regulatory urban background monitoring station (R801) in Antwerp, Belgium. Sensor systems were evenly distributed across the three shelter levels. Regulatory data included NO_2_ (Thermo 42C; µg/m^3^), O_3_ (Teledyne API400E; µg/m^3^), PM_1_, PM_2_._5_, PM_10_ (Palas FIDAS 200; µg/m^3^), BC (Thermo MAAP; µg/m^3^), relative humidity (%), and temperature (°C) and exhibited good hourly data coverage (n = 2132) of 96.7, 96.6, and 92.9% for, respectively, PM, BC, and NO_2_. The collected raw (RAW) and lab-calibrated (LAB CAL; linear calibration based on lack-of-fit) sensor data were subsequently evaluated for the following:Hourly data coverage (%)Timeseries plot: RAW & LAB CALScatter plot: RAW & LAB CALComparability between sensors: between-sensor uncertainty (BSU)Comparability with reference (hourly): R^2^, RMSE, MAE, MBEExpanded uncertainty (non-parametric): Uexp (%)

In addition we evaluated the sensitivity of the sensors (R^2^, RMSE, MAE, MBE) towards the (real-life) coarse particulate fraction (PM_10_–PM_2_._5_) and exhibited meteorological conditions (temperature and relative humidity). Moreover, we tested the impact of a 2-week field co-location calibration (FIELD CAL; linear calibration for PM and multilinear for NO_2_) on the resulting sensor performance and compared the field calibration performance to the lab calibration performance.

## 3. Results

### 3.1. Laboratory Test

#### 3.1.1. PM

Due to the varying monitoring resolutions of the sensor systems (2 s–5 min; Appendix A), all data were temporally aggregated to a 1-min resolution and merged with the reference (Grimm 11D) data. The SODAQ Air and NO_2_ apply a 5 min resolution when stationary and change automatically to ~10 s when mobile, resulting in fewer datapoints in the laboratory test. The GeoAir experienced power supply issues during the lack-of-fit measurements (insufficient amperage from applied USB hubs), resulting in data loss for all sensors (NA in Table 1). Setpoint averages (µg/m^3^) were calculated from the most stable concentration periods (final 15 min of each 1-h setpoint) and are shown in Appendix A. From these setpoint averages, lack-of-fit (linear regression) curves were generated (Appendix A), linearity (R^2^) and regression coefficients (slope + intercept (y = a*x + b) and slope only (y = a*x)) determined and sensor accuracy (%) were calculated. All results are shown per sensor system and subsequently presented in an overview table.

All sensor systems respond nicely to the increasing particle concentrations inside the PM exposure chamber (Figure 4), resulting in a generally good linearity between sensor and reference (R^2^ = 0.96–1). Nevertheless, most of the sensor systems seemed to underestimate the actual PM_2_._5_ and PM_10_ concentrations, while overestimating the PM_1_ particle size fraction. Mean setpoint accuracy (mean of different setpoint accuracies) varied from 82–85% for PM_1_, 63–69% for PM_2_._5_, and 28–31% for PM_10_ (ATMOTUBE); 12–28% for PM_1_, 76–84% for PM_2_._5_, and 45–51 for PM_10_ (TERA PMscan); 80–86% for PM_1_, 53–56% for PM_2_._5_, and 22–23 for PM_10_ (OPEN SENECA); 31–94% for PM_1_, 48–95% for PM_2_._5_, and 20–43 for PM_10_ (SODAQ Air); 60–77% for PM_1_, 35–70% for PM_2_._5_, and 13–29 for PM_10_ (SODAQ NO_2_); and 63% for PM_1_, 29% for PM_2_._5_, and 13% for PM_10_ (2BTECH PAM). Quantitative performance statistics are calculated based on all 1 min averaged lack-of-fit data (R^2^, MAE, BSU and Uexp) for each sensor system and particle size fraction and shown in Table 1. 

From Figure 4, it can be observed that the between-sensor uncertainty (BSU) is larger for the SODAQ Air (3.96 µg/m^3^) and NO_2_ (no simultaneous data) when compared to ATMOTUBE (1.52 µg/m^3^), OPEN SENECA (1.21 µg/m^3^), and TERA PM (1.64 µg/m^3^). For the 2BTech PAM, this could not be evaluated, as we had only one device available.

After applying a linear lab calibration (based on lack-of-fit regression coefficients), all sensor systems fell within expanded uncertainty <50% for PM_2_._5_, which is the data quality objective for indicative (Class 1) sensor systems (cfr. CEN/TS 17660-1 for gases).

Recent research showed that particle sensors exhibit low sensitivity in the coarse particle size range (2.5–10 µm) [43,44]. Therefore, a test procedure was developed to evaluate sensor sensitivity to the coarse fraction and to evaluate if sensors really measure PM_10_ rather than extrapolating it from the PM_2_._5_ signal. We exposed the sensors to monodisperse dust (silica microspheres) of, consecutively, 7.75 µm and 1.18 µm (fine) diameters. We finetuned the settings of the aerosolizer to reach representative (~100–150 µg/m^3^) PM_10_ concentrations by generating dust pulses every 30 s during a 5 min period. The idea is to simulate conditions with mainly fine (‘Fine test cond’.) and mainly coarse aerosol (‘Coarse test cond’.), respectively. Two representative 5-min periods (1 coarse test, 1 fine test) were subsequently selected and evaluated by calculating the dust composition (% coarse), PM_10_, PM_2_._5_, and PM_coarse_ sensor/REF ratios, and 2 relative change metrics as (3) and (4) (%): Relative change (%) in fractional (coarse vs. fine) sensor/REF ratio during respective fine and coarse test conditions:
(3)$${Rel}_{{PM}_{fractional}}\left(\%\right)=\frac{\left(\frac{{PM}_{10-2.5\;(sen,\;COARSE)}}{{PM}_{10-2.5\;(REF,\;COARSE)}}-\frac{{PM}_{2.5\;(sen,\;FINE)}}{{PM}_{2.5\;(REF,\;FINE)}}\right)}{\frac{{PM}_{2.5\;(sen,\;FINE)}}{{PM}_{2.5\;(REF,\;FINE)}}}\times 100$$

Relative change (%) in PM_10_ sensor/REF ratio between fine and coarse test conditions:


(4)
$${Rel}_{{PM}_{10}}\left(\%\right)=\frac{\left(\frac{{PM}_{10\;(sen,COARSE)}}{{PM}_{10\;(REF,\;COARSE)}}-\frac{{PM}_{10\;(sen,\;FINE)}}{{PM}_{10\;(REF,\;FINE)}}\right)}{\frac{{PM}_{10\;(sen,\;FINE)}}{{PM}_{10\;(REF,FINE)}}}\times 100$$


The sensor systems tend to visually pick up fine particle spikes but appeared far less responsive to the coarse fraction spikes (Figure 5). Note that in both fine and coarse generation spikes, PM_2_._5_ is present. Similar responses are observed between the different sensor systems, which is not surprising, as all sensors are ultimately based on three original equipment manufacturer (OEM) sensors, namely Sensirion SPS30, Plantower PMS, and TERA next-PM. From the calculated change ratios in Appendix A, the sensor/REF ratio changed significantly between the considered particle size conditions (73–100%), with all sensors showing very low sensitivity towards the coarse particle size fraction (PM_coarse_ sensor/REF ratio from 0–0.11 as shown in Appendix A).

#### 3.1.2. NO_2_

For all sensors containing a NO_2_ sensor (3/10), lack-of-fit tests were conducted on three days (August 12th, 14th, and 15th) at concentrations ramping between 0 and 200 µg/m^3^ (Figure 6). Due to the varying monitoring resolutions of the sensor systems (2 s–5 min), all data were temporally aggregated to 1-min resolution and merged with the reference data (Thermo NO_x_ analyzer). Setpoint averages were calculated based on steady-state conditions (final 1.5-h considering a 15-min buffer period before each setpoint change). From these setpoint averages, lack-of-fit (linear regression) plots were generated, linearity (R^2^) and regression coefficients (slope + intercept (y = a*x+b) and slope only (y = a*x)) determined and sensor stability (µg/m^3^) and accuracy (%) were calculated. The SODAQ NO_2_ showed significant noise and data connectivity issues, resulting in a low stability (5–80 µg/m^3^) and setpoint accuracy (−113–254%). Moreover, sensor readings were inversely correlated (R^2^ = 0.03–0.18) to the actual NO_2_ concentrations (Figure 6), with a poor between-sensor uncertainty (BSU) of 125 µg/m^3^. This out-of-the-box performance can be considered as inadequate. Potential calibration is hindered by the high signal noise, while sensor boxes showed connectivity issues and high BSU. The 2BTech PAM (only one unit available) was positively correlated with the generated NO_2_ concentrations, with a mean setpoint accuracy of 72%, but exhibited significant noise and extreme peak values during the lack-of-fit test, resulting in low sensor stability of 27 µg/m^3^. The DST Observair (one unit available) is not pre-calibrated by the supplier and relies on co-location calibration in the field. The uncalibrated sensor readings during the lack-of-fit test varied between -0.03 and 0.03 µg/m^3^ and showed a negative linear response to the increasing NO_2_ concentration steps. Compared to the SODAQ NO_2_ and PAM, the Observair exhibits much lower signal noise, resulting in better stability (<0.01 µg/m^3^) and better calibration potential. After calibration, the expanded uncertainty (Uexp) of the Observair (65%) outperforms the observed accuracies of the SODAQ NO_2_ (415–490%) and PAM (80%). Nevertheless, the considered NO_2_ sensors do not classify for the Class 1 uncertainty objective of <25% (CEN/TS 17660-1 [45]).

The impact from a changing relative humidity (0-50-75-90%) at zero and span concentration resulted in similar responses (Appendix A), with initial peak responses with every setpoint change followed by a subsequent stabilization (transient effect) under different levels of noisiness (Observair < PAM < SODAQ NO_2_). Similar responses can be explained by the underlying OEM sensor (Alphasense NO2-B43F), which is similar for all NO_2_ sensor systems. Similar transient effects (Appendix A) were observed under varying temperatures (−5, 10, 20, and 30 °C), both at zero and span concentration.

To evaluate response time to rapidly changing NO_2_ concentrations, sensors were placed in glass tubes that allowed for rapid concentration changes from 0–200 µg/m^3^ (Appendix A). The smaller volume of the glass tubes (compared to the NO_2_ exposure chamber) only allowed evaluation of the Observair and PAM sensors as the SODAQ NO_2_ boxes did not fit in the glass tubes. Thirty-min setpoints (0 and 200 µg/m^3^) were considered, and lab-calibrated sensor data were compared to the 1-min data from the Thermo NO_x_ analyzer. Averages and 90-percentiles (90% of max concentration) concentrations were determined for each 200 µg/m^3^ plateau, and the associated response time, i.e., time needed to reach 90% concentration, was calculated for each sensor system (and reference analyzer). The resulting response times derived from the 3 consecutive 0–200 plateaus are provided in Appendix A and varied from 1–2 min for the sensor systems and 3 min for the Thermo NO_x_ reference analyzer. Quantitative performance statistics (R^2^, MAE, BSU, and Uexp) are calculated based on all 1 min averaged lack-of-fit data for each sensor system and are shown in Table 1.

### 3.2. Mobile Field Test

All sensors were mounted on top (in the free airflow) of a cargo bike. Package sleeves were applied to damp vibrations of the cargo bike while cycling. Besides the sensors, two mid-range instruments, namely a Grimm 11D (PM; without heated inlet) and MA200 (BC), were placed inside the cargo bike with air inlets at the height of the sensors. Finally, the cargo bike was equipped with 3 different GPS instruments (Garmin 810 Edge, TomTom Runner 2, Komoot smartphone application). The TomTom track showed the highest monitoring resolution (1 s) and horizontal accuracy and was, therefore, selected as reference track. The exhibited PM_2_._5_ concentration variability (measured by the Grimm) ranged between 4.8 and 133.3 µg/m^3^, while the BC (measured by the MA200) varied between 0.4 and 4.4 µg/m^3^ (Appendix A). While the highest PM_2_._5_ concentrations were observed at a housing façade construction site, the highest BC concentrations were obtained when cycling downwind of a busy highway (E313/E34). When plotting all sensor tracks on a map (Figure 7), the GPS accuracy performed visually better in open areas compared to narrow and/or high street canyons. A higher height/width ratio seems to result in lower GPS accuracy, while GPS accuracy deteriorates as well when moving through tunnels, which are well-described phenomena in the literature [27,46,47].

When calculating the average horizontal accuracy (m) as average distance to the reference track in QGIS (Figure 7), the horizontal accuracy was generally good, achieving a <10 m horizontal accuracy for all sensor systems (Appendix A). The highest horizontal accuracy (2.28 m) was obtained for the TERA PMscan, while the lowest horizontal accuracy (8.15 m) was observed for the GeoAir.

With regard to the measured raw sensor signals (PM/NO_2_/BC), the mobile deployment (and related vibrations) did not seem to result in additional instrument noise or outliers when compared to stationary conditions. Moreover, similar hotspots were identified when comparing the sensor systems to the high-grade (MA200 and Grimm) monitors (Appendix A).

### 3.3. Field Co-Location Campaign

All sensor systems were deployed for 3 months (7 September 2022–5 December 2022) in an actively vented exposure shelter on top of an urban background monitoring station (R801) in the city center of Antwerp (Figure 8). Different data storage and transmission protocols were used, including automatic cloud upload via GPRS/4G (SODAQ) and internal SD card storage (GeoAir), while some sensor systems relied on a smartphone application (TERA PMscan, ATMOTUBE) or a combination of these data transmission protocols (PAM, OPEN SENECA, Airbeam, Observair). Some sensor systems were not designed for continuous, long-term monitoring. TERA PMscan relies on a smartphone application for operation, which resulted in forced automatic shutdowns by the smartphone software after some time (~1–2 days) and lack of continuous long-term data. The Observair relies on filter replacements for its BC measurement. As the filter saturates quickly, the instrument turned to error mode and did not collect any BC or NO_2_ data. The BCmeter also relies on filter replacements. A dedicated 1.5 week campaign (16–30 November) was therefore set up to evaluate BC (and NO_2_ from the Observair). The Airbeams arrived later and became operational on the 9th of November. Sensor data were offloaded (remotely via web dashboards and on-site via SD card readout) weekly to avoid data loss, and a logbook was created to keep track of that status and encountered issues. 

From the regulatory data, PM_2_._5_ concentrations ranged from 1–51 µg/m^3^ (mean = 10.85 µg/m^3^), while NO_2_ exhibited 2–111 µg/m^3^ (mean = 26 µg/m^3^). Atmospheric temperature varied between 1 and 27 °C (mean = 13 °C), while relative humidity was within 42 and 100% (mean = 83.5%). Temporal pollutant variability reflects typical urban pollution dynamics (Appendix A), with morning and evening rush hour peaks for NO_2_ and BC, slightly delayed PM peaks with a regional background character, and O_3_ that is photochemically formed at low NO_2_ concentrations and high solar radiation conditions (inversely related to NO_2_).

For each of the sensor systems, hourly data coverage, linearity (R^2^), accuracy, expanded uncertainty, impacts from lab and field calibration, and sensor drift (sensor/REF ratio) over time were evaluated (Table 2). For PM sensor systems, the sensitivity towards the coarse particle fraction (PM_10_–PM_2.5_) and impact from, respectively, lab- and field calibrations were additionally evaluated. PM field calibration was similar to the lab calibration, based on linear slope/intercept derivation based on a training period (first 2 weeks: 7 September 2022–21 September 2022) and evaluated (R^2^ and MAE) based on the remaining 2.5 months of data (22 September 2022–5 December 2022). For NO_2_ sensor systems, a multilinear field calibration model was trained with covariates for sensor response, temperature, RH, and O_3_, following earlier sensor calibration studies [48,49,50]. Model training was based on 2 weeks of co-location data (to fit the model and derive regression parameters), and the calibration performance (R^2^ and MAE) was tested on the remaining 2 months of test data. This multilinear field calibration outperformed the raw and lab calibrations for all NO_2_ sensor systems. Lab calibrations did not hold in field conditions, which is not surprising, as field conditions are different in terms of PM composition and meteorological conditions (temperature, relative humidity). Compared to the observed PM_2_._5_ performance in Table 2, performance decreases for PM_10_ (R^2^ = 0.6–0.62, MAE = 12.6 µg/m^3^), and the association is entirely lost (R^2^ = 0–0.01) when focusing on the coarse fraction (PM_coarse_ = PM_10_-PM_2.5_), confirming the lack of sensitivity in the coarse particle size fraction. This was also observed in an earlier field study with six different low-cost PM sensors [43]. For PM_2.5_, general good correlations (R^2^ = 0.7–0.9), varying accuracies (MAE = 3–4.7 µg/m^3^), and low between-sensor uncertainties (0.1–0.7 µg/m^3^) were observed. The accuracy worsened by applying the lab calibration but was optimized further for all sensor systems based on the field calibration. No distinct aging effect (gradual deviation in sensor/REF ratio) was observed over the 3-month co-location period. The considered PM sensor systems exhibited sensitivity towards relative humidity (Appendix A), with exponentially increasing sensor/REF ratios under increasing humidity (mainly impacting data quality from a relative humidity >85%). This phenomenon is caused by condensational particle growth due to particle hygroscopicity and is well documented in previous literature [43,48,51,52,53,54,55,56,57]. An overview of the observed quantitative performance metrics based on the hourly-averaged data for each of the sensor systems during the field co-location campaign is provided in Table 2.

Hourly PM_2.5_, NO_2_, and BC timeseries of the considered sensor systems and reference data are provided in Figure 9.

## 4. Discussion

During this lab and field benchmarking campaign, we collected quantitative and qualitative evidence on the fit-for-purpose of current commercially available dynamic exposure sensor systems. An overview is provided of the observed sensor system performance (hourly coverage, accuracy, R^2^, MAE, BSU, stability, Uexp) for the considered pollutants under laboratory (Table 1) and real-world (Table 2) conditions.

For the considered PM sensor systems, out-of-the-box performance is already quite good and close to the Class 1 data quality objective (U_exp_ < 50%). In addition, the sensors showed high precision, <0.4 µg/m^3^ in the lab and <0.6 µg/m^3^ in the field, which allows for multi-sensor (network) applications (e.g., [13,58,59]). Whether the obtained accuracy is sufficient to characterize PM gradients in urban environments (which are typically not that steep) will vary from city to city and should be further investigated. In our mobile field test, Grimm measurements showed PM_2_._5_ concentrations along the 10km trajectory ranging from 4.8 to 133 µg/m^3^. This exposure variability is, therefore, quantifiable by the considered sensor systems with MAEs of 3–4.7 µg/m^3^. The highest accuracy was observed for PM_1_, followed by PM_2_._5_ and PM_10_. The considered sensor systems do not reliably detect the coarse particle size fraction or show sensitivity towards relative humidity. TERA is the only sensor system that seems to pick up some coarse particles (R^2^ = 0.3), while all other sensors show R^2^ of ~0. The accuracy of PM sensors can be further improved by linear slope/intercept calibration. However, we showed that lab calibrations do not hold in the field, as previously shown in other studies [11,13,48,60]. A local field calibration (representative pollutant and meteorological environment) seems, therefore, crucial to obtain the most reliable sensor data. In general, the assessed PM performance and observed sensitivities (drift/RH) are very similar between the benchmarked PM sensors, which can be explained by similar underlying sensor technology (Sensirion SPS30 + Plantower) and lack of applied factory algorithms. The sensor systems showed elevated sensor/REF ratios under increasing relative humidity, which can be explained by hygroscopic effects documented in previous literature [48,51,52,53,54,55,57,61,62].

Regarding NO_2_, out-of-the-box performance was unsatisfactory for direct application, as sensor systems suffered from noise (stability) and calibration (negative association) issues. Although 2BTech PAM showed the best raw performance, a higher but negative association (R^2^) and stability were observed for Observair. Following a linear laboratory calibration, the best performance was, therefore, achieved for the Observair. Similar to the PM sensors, linear lab calibrations do not hold in the field. For NO_2_, a local and multilinear field calibration (incorporating covariates for temperature, relative humidity, and O_3_ sensitivity) showed acceptable sensor performance (R^2^ = 0.75–0.83, MAE = 6–44 µg/m^3^), which shows the potential of the considered NO_2_ sensor systems. Further research and development work should therefore focus on implementing research-proven noise reduction and calibration procedures [11,48,49,60,63,64,65] in commercial instruments to increase the level of maturity on the market. Recent sensor studies applying multilinear [48,60] or machine-learning-based [64,66] calibrations (co-location or network-based) have provided evidence on sensor sensitivities and data quality improvements on a variety of sensors. Application of this knowledge to commercial applications is crucial in order to obtain reliable and actionable air quality data. 

Regarding BC, both considered sensor systems showed good field performance (R^2^ = 0.82–0.83, MAE = 0.2–0.3 µg/m^3^); however, we should mention that BCmeter cannot yet be applied in mobile applications (due to wired power and wifi connectivity). The measurement principle of light attenuation on filter strips has proven to be a robust methodology to measure black carbon in the past [15,67,68,69,70,71,72] and can be minimized to portable and lower-cost instruments. Moreover, the spatial BC exposure variability, measured by the Observair in the mobile field test, was in good agreement with the Aethlabs MA200 measurements (Appendix A). In general, all sensor systems showed a good horizontal accuracy (<10 m) with no vibration impacts on the sensor readings for all pollutants during the mobile field test, confirming the suitability of portable sensor systems for mobile applications.

## 5. Conclusions

This study evaluated the fit-for-purpose of commercially available portable sensor systems for dynamic exposure assessments in urban environments. We evaluated 10 sensor systems, measuring PM, NO_2_, and/or BC in both laboratory and real-world conditions. Besides quantitative performance assessments, qualitative experience on their portability, data transmission/storage, and user-friendliness were obtained throughout the experiments. Autonomous operation with internal GPS (no reliance on app connectivity) and data storage redundancy (SD storage besides cloud or app transmission) for example showed to be valuable assets in terms of data coverage. Results of the considered sensor systems indicate that out-of-the-box performance is relatively good for PM and BC, but the maturity of the tested NO_2_ sensors is still low, and additional effort is needed in terms of signal noise and calibration. Multivariate calibration under field conditions showed promising potential for real-world applications. Future directions for PM and BC should focus on applicability (pollutant gradients in urban environments), added value, and user-friendliness (day-to-day use) of real-world applications, while for NO_2_, research-proven noise reduction and calibration procedures [11,48,49,60,63,64,65] should be implemented in commercial instruments to increase the level of maturity on the market. This work shows that commercially available portable sensor systems have reached a good maturity level for PM and BC, while more work is needed for NO_2_ in terms of calibration and noise reduction. More accurate and dynamic exposure assessments in contemporary urban environments are crucial to study real-world exposure of individuals and the impact on potential health endpoints [17,73,74,75,76,77,78]. This research domain will be boosted by the greater availability of mobile monitoring systems capable of quantifying urban pollutant gradients and enabling personal exposure assessments, identification of hotspot locations, and new air quality mapping applications, in turn driving awareness, behavior change, and evidence-based air quality policies.

## Figures and Tables

**Figure 1 sensors-24-05653-f001:**
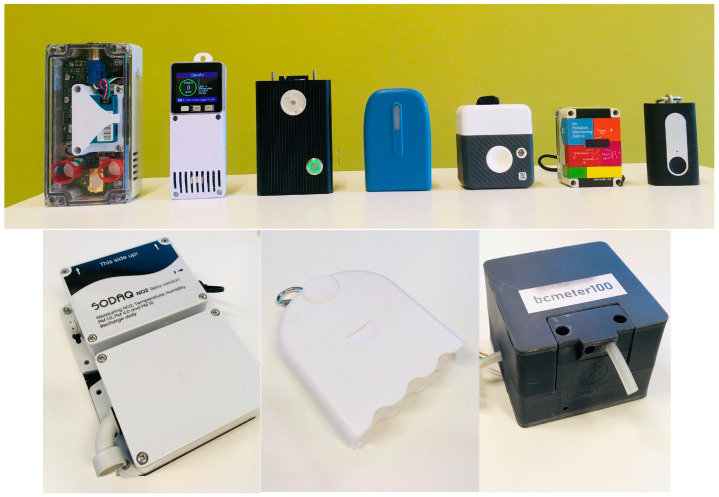
Considered sensor systems (10) with (upper panel left to right): PAM (2BTech, Broomfield, CO, USA), GeoAir, Observair (DSTech, Pohang-si, Republic of Korea), SODAQ Air (SODAQ, Hilversum, The Netherlands), PMscan (TERA Sensor, Rousset, France), OPEN SENECA (open-seneca.org), and ATMOTUBE Pro (ATMOTECH Inc., San Francisco, CA, USA). Lower panel left to right: SODAQ NO_2_ (SODAQ, Hilversum, The Netherlands), Habitatmap Airbeam (Habitatmap, Brooklyn, NY, USA), and BCmeter (BCmeter.org).

**Figure 2 sensors-24-05653-f002:**
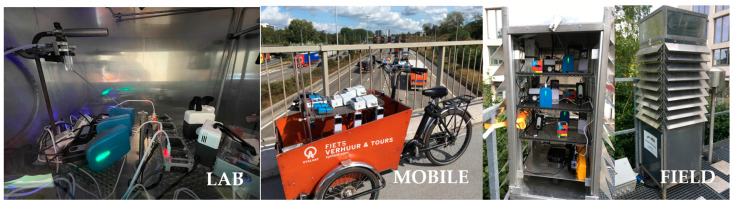
PM exposure chamber in the lab (**left**), mobile field test with cargo bike (**middle**), and field co-location campaign at an urban background monitoring station (**right**).

**Figure 3 sensors-24-05653-f003:**
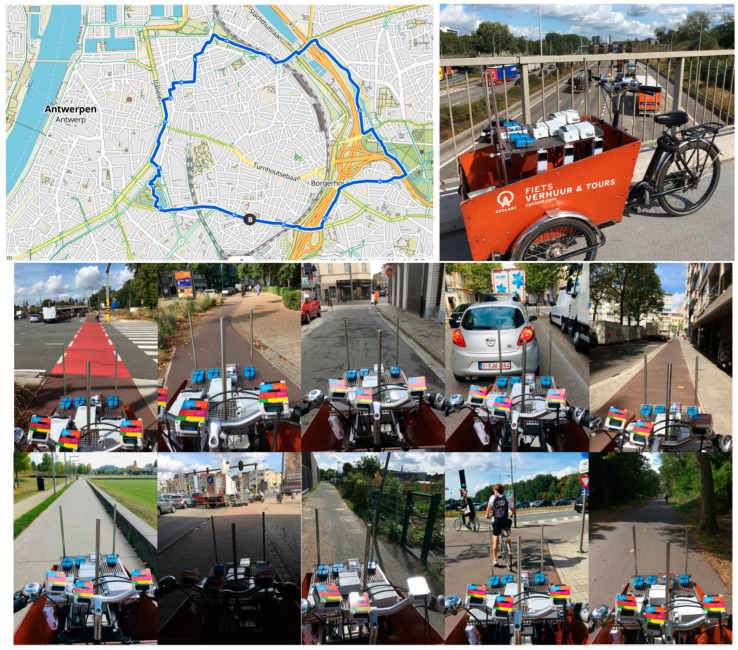
Mobile field trajectory (10.4 km) in the city center of Antwerp, Belgium (**upper left**), and applied cargo bike setup (**upper right**). Lower pictures show the variety of urban landscape and road traffic along the cycling route.

**Figure 4 sensors-24-05653-f004:**
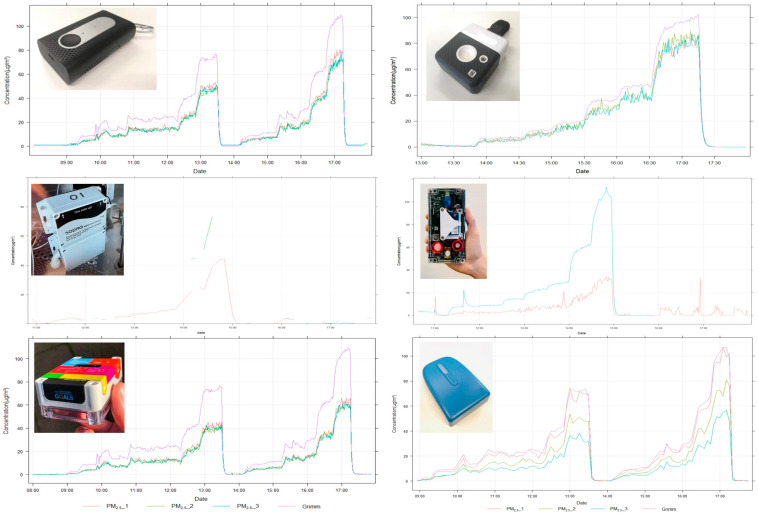
Stepwise PM_2_._5_ concentrations generated during the lack-of-fit test and measured concentrations by the different sensor systems (1-3; green-blue-red) and the reference monitor (Grimm; purple/green).

**Figure 5 sensors-24-05653-f005:**
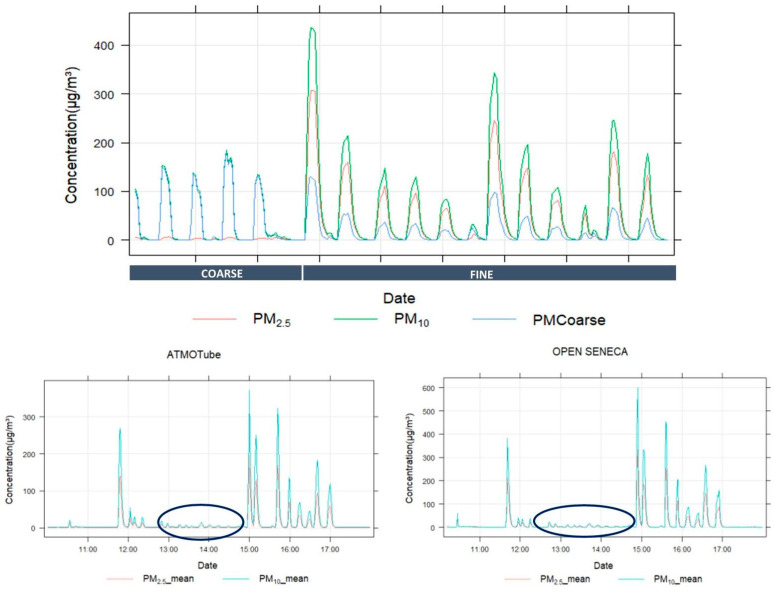
Coarse PM testing procedure with consecutive 5-min generation periods of coarse (7.75 µm) and fine (1.18 µm) PM peaks (upper panel; measured by Grimm REF monitor) and resulting ATMOTUBE and OPEN SENECA sensor response (µg/m^3^) in the lower panels.

**Figure 6 sensors-24-05653-f006:**
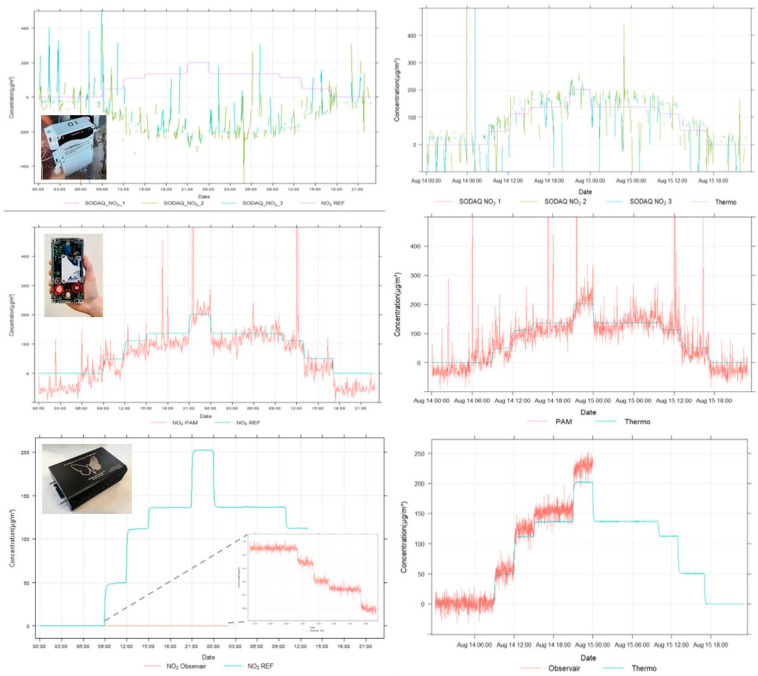
Stepwise NO_2_ concentrations generated during the lack-of-fit tests and measured raw (**left**) and lab-calibrated (**right**) concentrations by the SODAQ NO_2_ (1-3; upper in red), PAM (**middle** in red), Observair (**lower** in red), and the reference monitor (Thermo NO_x_ analyzer in purple/green).

**Figure 7 sensors-24-05653-f007:**
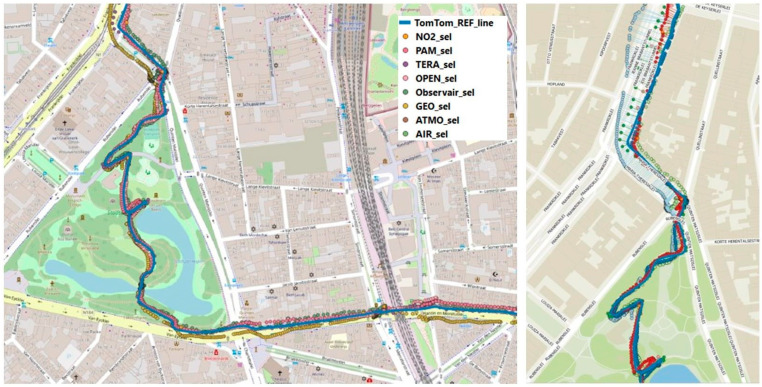
(**Left**): GPS tracks of the considered sensor systems (dots) and reference GPS track (blue line). (**Right**): Accuracy calculation by means of horizontal distance to reference GPS track (blue line).

**Figure 8 sensors-24-05653-f008:**
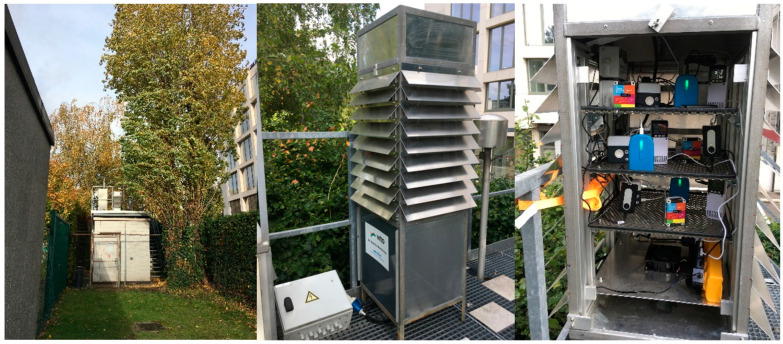
Location of the exposure shelter on top of R801 urban background monitoring station (**left**), detail of the exposure shelter (**middle**), and positioning of the sensor systems at the different platforms inside the shelter (**right**).

**Figure 9 sensors-24-05653-f009:**
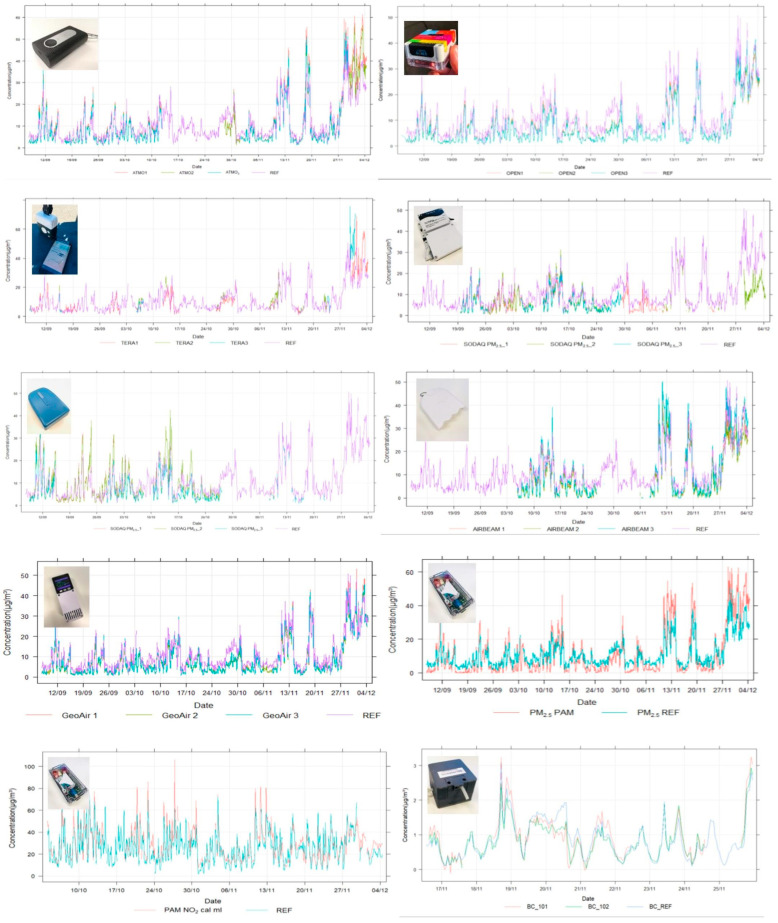
Hourly timeseries of PM_2_._5_, NO_2_, and BC concentrations measured by the respective sensor systems and the reference monitors at the R801 reference background monitoring station.

**Table 1 sensors-24-05653-t001:** Summary table of out-of-the-box performance (setpoint accuracy, setpoint stability, MAE, R^2^, Uexp, and BSU) obtained for each considered sensor system and pollutant (PM and NO_2_) during the laboratory tests.

	SENSOR SYSTEM	Accuracy (%)			MAE	R^2^	Uexp	BSU
		PM_1_	PM_2.5_	PM_10_	µg/m^3^	-	%	µg/m^3^
PM	ATMOTUBE (3)	84	65	29	10.0	0.98	47	1.5
	OPEN SENECA (3)	83	54	22	12.6	0.99	55	1.2
	TERA (3)	18	79	47	5.2	1.00	25	1.6
	SODAQ Air (3)	64	70	31	8.9	0.99	40	4.0
	SODAQ NO_2_ (3)	68	52	21	10.9	0.99	45	NA
	GeoAir (3)	NA	NA	NA	NA	NA	NA	NA
	PAM (1)	63	29	13	17.3	0.96	79	NA
	**SENSOR SYSTEM**	**Accuracy**	**Stability**		**MAE**	**R^2^**	**Uexp**	**BSU**
		**%**	**µg/m^3^**		**µg/m^3^**	**-**	**%**	**µg/m^3^**
NO_2_	SODAQ NO_2_ (3)	−166	51		270.3	0.11	304	124.7
	PAM (1)	72	27		49.5	0.13	110	NA
	Observair (1)	0	0		79.0	0.98	112	NA

**Table 2 sensors-24-05653-t002:** Summary table of quantitative performance metrics (accuracy, stability, MAE, R^2^, Uexp, and BSU) obtained for each sensor system and pollutant (PM and NO_2_) during the field co-location campaign (hourly data). * As the PAM only consisted of one instrument, BSU could not be calculated (NA).

	SENSOR SYSTEM	Data Coverage	MAE	R^2^	Uexp	BSU
		%	µg/m^3^	-	%	µg/m^3^
PM_2.5_	ATMOTUBE (3)	76	4.3	0.88	48	0.6
	OPEN SENECA (3)	100	3.7	0.90	35	0.3
	TERA (3)	17	4.4	0.87	64	0.1
	SODAQ Air (3)	44	3.1	0.68	16	0.7
	SODAQ NO_2_ (3)	44	3.8	0.67	40	0.4
	AIRBEAM (3)	53	3.9	0.87	36	0.7
	GeoAir (3)	96	3.0	0.89	28	0.6
	PAM (1)	100	4.7	0.89	66	NA *
NO_2_	SODAQ NO_2__raw (3)	44	190.3	0.42	614	
	SODAQ NO_2__cal (1)	44	27.1	0.62	108	
	SODAQ NO_2__mlcal (1)	44	5.6	0.83	37	
	PAM (3)	100	84.1	0.55	284	
	PAM_cal (1)	100	349.0	0.55	1225	
	PAM_calml (1)	100	44.2	0.75	44	
	Observair_raw	78	28.4	0.38	111	
	Observair_cal	78	28.8	0.38	95	
	Observair_mlcal	78	NA	NA	NA	
BC	Observair	78	0.3	0.82		
	BCmeter	78	0.2	0.83		

## Data Availability

The research data of this study will be made available on request.

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
