# Peer review of "Portable Sensors for Dynamic Exposure Assessments in Urban Environments: State of the Science"

_sensors, 2024, doi:10.3390/s24175653_

Round 1
Reviewer 1 Report
Comments and Suggestions for Authors
Review Report
Title: Portable Sensors for Dynamic Exposure Assessments in Urban Environments: State of the Science
Authors: Jelle Hofman . Borislav Lazarov , Christophe Stroobants , Evelyne Elst , Inge Smets and Martine Van Poppel.
Keywords: air quality; sensors; exposure; assessment; citizens; validation
Summary of the Article
This article provides an extensive evaluation of commercially available portable sensor systems designed to measure particulate matter (PM), nitrogen dioxide (NO₂), and black carbon (BC) in urban environments. The goal of the research is to identify sensor systems suitable for mobile monitoring by citizens to assess dynamic exposure to air pollutants. The study involves both laboratory and field testing, including a mobile field test to evaluate GPS accuracy and sensor performance under real-world conditions. The results suggest that while PM and BC sensors show promising out-of-the-box performance, NO₂ sensors still require significant improvements.
二、Major Comments
1. Methodological Rigor
• The selection criteria for the 10 sensor systems could be more transparent. Providing additional details on how the final selection was made from the initial 39 candidates would enhance the clarity and replicability of the study.2. Data Presentation and Analysis
• While the data presentation is thorough, some sections, particularly the statistical analysis of sensor performance, could benefit from more concise and clearer explanations to improve readability and comprehension for a broader audience.3. Findings and Implications
• The findings indicate that while portable sensors for PM and BC are relatively mature, NO₂ sensors still face challenges such as signal noise and calibration issues. This distinction is critical and should be emphasized more clearly in the discussion to underline the specific needs for further development in NO₂ sensing technology. • The potential impact of mobile monitoring on public health policy and individual behavior is discussed but could be expanded to include more concrete examples or case studies illustrating the real-world benefits and applications of the sensors.Minor Comments
1. Abstract and Introduction
• The abstract effectively summarizes the study, but it would benefit from including specific findings or data points to give readers a clearer sense of the outcomes. • The introduction provides a solid background, but the review of existing literature could be more succinct to avoid redundancy and maintain focus on the unique contributions of the current study.
Direct_and_spillover_effects_of_new-type_urbanization_on_CO2_emissions_from_central_heating_sector_and_EKC_analyses_Evidence_from_144_cities_in_China/stats?pli=1&loginT=NVyovyIKe7OFWkTOIgiA1PqRp-NOsnjw_YjUUwUO6VypORz1wLYun-vwsmumoGstcgnxOlmAOSB160yY_e_qWu_gbSs&uid=Szis545z0wR1MznywGbpre423_x&
2. Materials and Methods
• The description of the sensor systems and benchmarking protocol is detailed, but including a flowchart or summary table outlining the main steps and tests could improve clarity and help readers quickly grasp the methodology. • The criteria for evaluating sensor accuracy, such as the choice of PM2.5 and PM10 concentration ranges, should be explicitly justified to demonstrate their relevance to real-world conditions3. Results and Discussion
• The results section is comprehensive, but some parts are overly technical and may benefit from simplification or additional explanations for readers less familiar with the subject matter. • The discussion could integrate more direct comparisons with previous studies to highlight advancements or differences in findings and to place the results within a broader research context.4. Figures and Tables
• The figures and tables are generally clear and informative. However, some graphs, particularly those depicting sensor performance across various conditions, would benefit from more distinct labeling or legends to ensure they are easily interpretable. • Consider including a summary table that consolidates the key findings for each sensor system to provide a quick reference for readers.5. Conclusion
• The conclusion effectively summarizes the main findings and implications of the study but could be enhanced by outlining specific recommendations for future research or development in the field of portable air quality sensors. • A brief discussion on potential applications and the broader impact of the findings on public health and policy would strengthen the conclusion. Recommendations for Improvement1. Clarify Selection Criteria
Provide a more detailed explanation of the selection process for the final 10 sensor systems, including specific criteria and decision-making steps.
2.Expand Discussion on NO₂ Sensors
Emphasize the challenges and potential solutions for improving NO₂ sensor performance, and discuss how advancements in this area could impact dynamic exposure assessments.
3.Simplify Data Presentation
Streamline the presentation of results and statistical analyses to make the information more accessible and easier to understand for a wider audience.
4.Highlight Real-World Applications
Include examples or case studies demonstrating how the findings can be applied in real-world scenarios to enhance public health and policy decisions.
5.Enhance Figures and Tables
Improve the clarity of figures and tables by adding more detailed legends, labels, and a summary table of key findings for quick reference.
Comments on the Quality of English Languagefurther revise
Reviewer 2 Report
Comments and Suggestions for Authors
General comments:
This study has the potential to provide useful personal sensor evaluation, since there are several newer sensors on the market. The between-sensor data are useful, as are the calculations for MAE. An appropriate reference sensor is used for both pollutants, and the plan to evaluate the samplers was sensible from some perspectives (range of concentrations, laboratory and field evaluation). However, humidity is a test parameter for NO2 but not PM. PM sensors are well-known to be sensitive to humidity. There is a wealth of literature that has demonstrated this, and the omission of this test is a fatal flaw. An evaluation of the PM sensors over a range of humidity levels needs to be conducted for the paper to be acceptable. A lesser issue is that the stated purpose of the tests is for use in personal sampling. It might have been more realistic to test the sensors on people during walks rather than on a bike, because the electronic response time might need to be faster when placed on a bike.
Minor comments:
Line 75: “Long list” is two words, not one.
Line 127: Please clarify the terms in the equation. Use C for concentration, define i and j. Should BSU_sensor be BSU_i?
Reviewer 3 Report
Comments and Suggestions for Authors
The manuscript has no specific technical breakthroughs, and cannot be treated as an article.
In particular, the goal should get more focused on promoting citizen sciences.
The spatial location of acquiring a <10 m horizontal accuracy for all sensor systems is not mentioned.
The research goals are not clearly mentioned in the Introduction.
The calibration of low-cost sensor is not provided or discussed, thus the reliability of datasets is a bit problematic in overall.
The application of Thermo Scientific 42iQ-TL monitor is not justified in a scientific basis.
There is a lack of explanation of why linear regression is used in various places or contexts within the manuscript.
Moreover, the authors should clearly highlight the scientific contribution made within current study, otherwise it cannot be published as an "article" format.
Comments on the Quality of English Language
Many different grammatical mistakes are detected, and the sub-script or under-script has not been handled properly.
Round 2
Reviewer 1 Report
Comments and Suggestions for Authors
I suggest accept
Comments on the Quality of English Languagefine
Author Response
Dear reviewer,
Thank you for the review and final approval.
On behalf of all co-authors,
Jelle Hofman
Reviewer 3 Report
Comments and Suggestions for Authors
The authors have tried their best to enhance the quality of the manuscript, and the quality of the manuscript is better now, with some scientific novelty and contribution from citizen science perspectives. Nevertheless, there are some weaknesses of the current manuscript:
(1) For Section 2.2.3: Field co-location campaign, please describe more technical specifications and details of the campaign and instruments installed on board
(2) The curves in Figures 4, 5, 6 and 9 should be of higher clarity, for example, being bolded. Currently, it's not obvious enough to be seen.
(3) For citizen science of portable environmental sensors, as well as how relevant data could contribute to promoting citizen science, please refer to the following references:
Section 4.1 and 4.2 (especially Section 4.2.3) of https://www.sciencedirect.com/science/article/abs/pii/S221067072100158X
https://www.frontiersin.org/journals/environmental-science/articles/10.3389/fenvs.2022.1019628/full
https://www.sciencedirect.com/science/article/pii/S2468067219300203
The Conclusion/ Discussion section should re-emphasize the importance of data utilization in enhancing citizen science in the future.
(4) Previous comment: The application of Thermo Scientific 42iQ-TL monitor is not justified in a scientific basis, please add in some references showing that Thermo Scientific monitor can give reliable results, that would be sufficient.
(5) For regression, normally one should apply y = ax+b, we should not assume that the y-intercept = 0. Please remove this possibility from relevant places.
(6) Where is the research gap that the study has filled in? It's hard to be visualized at this stage.
(7) In the last line of Introduction: "Doing so, we evaluate the applicability of these sensor systems in real-world urban environments" - there should be some environmental or geographical assumption in your study, so the results obtained in your case study might not be extended to other places. Please carefully elaborate how current results could be related to future research projects.
Another round of revision might be needed.
Comments on the Quality of English Language
A proper round of grammatical check is recommended. Some grammatical errors still appear.
Author Response
The authors have tried their best to enhance the quality of the manuscript, and the quality of the manuscript is better now, with some scientific novelty and contribution from citizen science perspectives. Nevertheless, there are some weaknesses of the current manuscript:
Thank you. We responded to all comments below in italic.
(1) For Section 2.2.3: Field co-location campaign, please describe more technical specifications and details of the campaign and instruments installed on board
Response 1: We agree. More technical details on instrumentation and setup were provided in the Results section, but now moved to the Materials & Methods section (Lines 192-200).
(2) The curves in Figures 4, 5, 6 and 9 should be of higher clarity, for example, being bolded. Currently, it's not obvious enough to be seen.
Response 2: We agree and enlarged Figures 4, 5, 6 and 9 in the revised manuscript to improve readability of the lines and legends.
(3) For citizen science of portable environmental sensors, as well as how relevant data could contribute to promoting citizen science, please refer to the following references:
Section 4.1 and 4.2 (especially Section 4.2.3) of https://www.sciencedirect.com/science/article/abs/pii/S221067072100158X
https://www.frontiersin.org/journals/environmental-science/articles/10.3389/fenvs.2022.1019628/full
https://www.sciencedirect.com/science/article/pii/S2468067219300203
The Conclusion/ Discussion section should re-emphasize the importance of data utilization in enhancing citizen science in the future.
Response 3: The focus of our study is not on citizen science, but on the technical readiness level of commercially available portable air quality sensors (e.g. for NO2). Nonetheless, citizen science can be an important field of application for low-cost (portable) sensors. We cited the aforementioned studies, together with other relevant CS references to stress the application potential of these sensor systems in the discussion and conclusion sections (e.g. at lines 538-544).
(4) Previous comment: The application of Thermo Scientific 42iQ-TL monitor is not justified in a scientific basis, please add in some references showing that Thermo Scientific monitor can give reliable results, that would be sufficient.
Response 4: See https://19january2021snapshot.epa.gov/sites/static/files/2019-08/documents/designated_reference_and-equivalent_methods.pdf --> p53. Moreover, In Belgium (Flanders) chemiluminescence (via Thermo Scientific 42iQ-TL monitor) is the regulatory reference/equivalent method to automatically measure NOx: https://www.vmm.be/data/alle-polluenten-actuele-waarden/methodiek-nox-meting. This is, therefore, the instrumentation we have available to compare the considered sensor systems against in field conditions.
(5) For regression, normally one should apply y = ax+b, we should not assume that the y-intercept = 0. Please remove this possibility from relevant places.
Response 5: When measuring the same variables and when no baseline offset is observed between the instruments/variables, linear calibration based on a single calibration factor (slope) is possible. Past studies based on independent training and test datasets showed that this calibration potentially improves data quality (we evaluated different linear approaches from a single to multiple covariates; slope, regression, multilinear):
- https://www.sciencedirect.com/science/article/abs/pii/S1309104221003093
- https://www.mdpi.com/2073-4433/13/6/944
- https://www.frontiersin.org/journals/environmental-health/articles/10.3389/fenvh.2023.1232867/full
(6) Where is the research gap that the study has filled in? It's hard to be visualized at this stage.
Response 6: As outlined in the introduction, a lot of literature exists on lab of field evaluations of commercial (stationary) low-cost sensors. Recent sensor applications are however focusing on network-based (multiple) dynamic (mobile) sensor applications and new pollutants (NO2 and BC). Such applications not only require good comparability against the reference, but different hardware (high monitoring resolution, GPS, form factor, battery) and data quality requirements (e.g. GPS accuracy, comparability against reference, low between-sensor uncertainty). The aim of our study was, therefore, to evaluate the fit-for-purpose of commercial portable air quality sensors; (i) quantitative evaluation of data quality within their application domain (mobile, urban environment, between-sensor-uncertainty vs observed pollutant gradient), (ii) qualitative evaluation of user friendliness, data storage, app functionality,... for available relevant anthropogenic urban pollutants; PM, NO2 and BC. In other words; to which end can commercial portable sensor solutions be applied in real-world applications; for PM and BC we showed that portable sensor solutions can readily be applied and further research should focus on data use, user friendliness, added value of mobile monitoring. For NO2, TRL levels are much lower and more work is needed on implementing research-proven noise reduction and calibration algorithms in commercial solutions.
(7) In the last line of Introduction: "Doing so, we evaluate the applicability of these sensor systems in real-world urban environments" - there should be some environmental or geographical assumption in your study, so the results obtained in your case study might not be extended to other places. Please carefully elaborate how current results could be related to future research projects.
Response 7: We agree and further constrained the geographical area to the considered urban environment at Lines 68-70: "Doing so, we evaluate the applicability of these sensor systems in a typical temperate Western-European and medium-sized (~200 km² and 2600 inhabitants/km²) urban environment of Antwerp, Belgium." Moreover, we clarified that pollutant gradients will be steeper in larger or more densely-populated cities (including relevant references).
Another round of revision might be needed.
Comments on the Quality of English Language
A proper round of grammatical check is recommended. Some grammatical errors still appear.
We made another grammatical check + an English-speaking colleague has read the manuscript as well and revised where necessary.
We think the manuscript has improved further and hope it can be reconsidered for publication in Sensors.